### A global satellite environmental data record derived from AMSR-E and AMSR2 microwave earth observations

Jinyang Du<sup>1</sup>, John S. Kimball<sup>1</sup>, Lucas A. Jones<sup>1</sup>, Youngwook Kim<sup>1</sup>, Joseph Glassy<sup>2</sup>, and Jennifer D. Watts<sup>1</sup>

<sup>1</sup> Numerical Terradynamic Simulation Group, College of Forestry & Conservation, The University of Montana, Missoula, MT 59812, United States <sup>2</sup>Lupine Logic, Inc., Missoula, MT 59802, United States

Correspondence to: Jinyang Du (jinyang.du@ntsg.umt.edu)

- Abstract. Space-borne microwave remote sensing is widely used to monitor global environmental changes for 10 understanding hydrological, ecological and climate processes. A new global land parameter data record (LPDR) was generated using similar calibrated, multi-frequency brightness temperature ( $T_b$ ) retrievals from the Advanced Microwave Scanning Radiometer for EOS (AMSR-E) and the Advanced Microwave Scanning Radiometer 2 (AMSR2). The resulting LPDR provides a long-term (Jun. 2002 - Dec. 2015) global record of key environmental observations at 25-km grid cell resolution, including surface fractional open water ( $f_w$ ) cover, atmosphere precipitable water vapor (PWV), daily maximum
- and minimum surface air temperatures ( $T_{mx}$  and  $T_{mn}$ ), vegetation optical depth (*VOD*) and surface volumetric soil moisture (*vsm*). Global mapping of the land parameter climatology means and seasonal variability over the full-year records from AMSR-E (2003-2010) and AMSR2 (2013-2015) observation periods is consistent with characteristic global climate and vegetation patterns. Quantitative comparisons with independent observations indicated favorable LPDR performance for *fw* ( $R \ge 0.75$ ; RMSE \le 0.06), *PWV* ( $R \ge 0.91$ ; RMSE \le 4.99 mm),  $T_{mx}$  and  $T_{mn}$  ( $R \ge 0.90$ ; RMSE  $\le 3.48$  °C), and *vsm* (0.63  $\le R \le$
- 0.84; bias corrected RMSE  $\leq$  0.06 cm<sup>3</sup>/cm<sup>3</sup>). The LPDR derived global *VOD* record is also proportional to satellite observed NDVI (GIMMS3g) seasonality ( $R \geq$  0.88) due to synergy between canopy biomass structure and photosynthetic greenness. Statistical analysis shows overall LPDR consistency, but with small biases between AMSR-E and AMSR2 retrievals that should be considered when evaluating long-term environmental trends. The resulting LPDR and potential updates from continuing AMSR2 operations provide for effective global monitoring of environmental parameters related to vegetation
- activity, terrestrial water storage and mobility; and are suitable for climate and ecosystem studies. The LPDR data set is publicly available at http://files.ntsg.umt.edu/data/LPDR\_v2.

#### **1** Introduction

Earth's atmospheric, biophysical and hydrological processes are closely coupled (Walko et al., 2000; Trenberth et al., 2007) and respond to altered climate forcing manifested by changes in key environmental variables (Meehl et al., 2007). Integrated

and consistent measurements of Earth System environmental variables at global scale are needed for advancing our

# Science Science Science

understanding of interconnected Earth systems (Trenberth et al., 2007) and for addressing critical global change related questions including global water cycle intensification (Huntington et al., 2006; Wild et al., 2008; D éry et al., 2009), arctic amplification and feedbacks to climate change (Smith et al., 2005; Grosse et al., 2011) and the primary drivers behind global vegetation changes (Zhu et al., 2016).

- Complementary to optical-thermal infrared (IR) and active microwave remote sensing, space-borne passive microwave radiometers allow for measurements of global environmental variables at relatively coarse spatial resolution (~ 5km to 100 km) but with relatively high temporal fidelity (~daily for higher latitudes ≥ 45°N) and with reduced constraints from variable solar illumination, clouds and atmosphere aerosol contamination effects (Ulaby et al., 2014). While lower frequency (e.g. L-band) sensors, including the ESA Soil Moisture and Ocean Salinity (SMOS) and NASA Soil Moisture Active-Passive
- (SMAP) missions, are generally considered optimal for detecting soil and surface water signals under moderate to high vegetation biomass conditions (Kerr et al., 2001; Entekhabi et al., 2010), higher frequency sensors, such as AMSR-E (Koike et al., 2004) and AMSR2 (Imaoka et al., 2012), provide simultaneous multi-channel (C- to W-band)  $T_b$  observations with variable sensitivity to surface water, soil, vegetation and atmosphere conditions (Njoku et al., 2003; Jones et al., 2010); the combined observations allow for distinguishing individual land parameter signals from background noise. However, the
- major AMSR-E and AMSR2 (hereafter denoted as AMSR-E/2) algorithms have largely focused on single parameter retrievals, including the NASA and JAXA standard soil moisture products (Njoku et al., 2003, Koike et al., 2004). In contrast, the University of Montana (UMT) global Land Parameter Data Record version 1 (LPDR v1) was developed to exploit AMSR-E multi-frequency  $T_b$  observations for global daily mapping of multiple synergistic land parameters related to the status and storage of water in the atmosphere, vegetation and soil (Jones et al., 2010). The LPDR v1 database has been
- applied for a variety of environmental studies, including quantifying surface water inundation impacts on tundra methane emissions (Watts et al., 2014); boreal wildfire disturbance and recovery assessments (Jones et al., 2013); evaluating hydroclimatic controls on vegetation phenology (Alemu and Henebry, 2013; Guan et al., 2014); biodiversity modeling and prediction (Waltari et al., 2014); and vector borne disease risk assessments (Chuang et al., 2012). The LPDR v1 has also served as a baseline for evaluating other AMSR-E algorithm retrievals (Mladenova et al., 2014) and refinements (Jang et al.,
- 2014; Du et al., 2014). The LPDR v1 encompasses the AMSR-E record (2002-2011), while similar observations from AMSR2 enable potential LPDR continuity (Du et al., 2014).

In this investigation, the Version 2.0 UMT Land Parameter Data Record (henceforth denoted as LPDR) was generated by incorporating recent algorithm improvements (Du et al., 2015; Du et al., 2016a), new algorithm refinements and an extended AMSR-E/2 satellite record. The key satellite microwave land parameter retrievals derived from this study include daily

maximum and minimum surface air temperature ( $T_{mx}$  and  $T_{mn}$ ), atmosphere precipitable water vapor (*PWV*), vegetation optical depth (*VOD*), surface fractional open water cover (*fw*) and volumetric soil moisture (*vsm*). Surface air temperature, defined as air temperature at approximate 2-meter height in this study and used as a global warming indicator (Hansen and Lebedeff, 1987; Jones et al., 1999), integrates key information on the thermal state of the land-atmosphere interface (Jones et al., 2010). *PWV* represents the total water content of the atmosphere column within the satellite sensor field-of-view (Bedka

et al., 2010), and is strongly interactive with temperature and climate (Held and Soden, 2000; Wentz et al., 2007). The *VOD* parameter represents the slant-path opacity of the intervening vegetation layer to land surface microwave emissions; *VOD* is microwave frequency dependent and is sensitive to changes in canopy biomass water content, including woody and foliar elements (Shi et al., 2008; Jones et al., 2011; Liu et al., 2011). The *fw* parameter is an important hydrological and biogeochemical variable (Watts et al., 2012), while large-scale mapping of *fw* dynamics has been used for studying high-latitude ecosystems, wetlands and carbon cycle related feedbacks to climate change (Van Huissteden et al., 2011; McVicar et al., 2012; Lupascu et al., 2014). Another key parameter is surface soil moisture, which governs the exchanges of water, energy and carbon between the soil and atmosphere (Entekhabi et al., 2010); soil moisture is defined in this study as the volume of water in a given volume of soil. The relative depth of soil moisture sensitivity is dependent on microwave frequency and land surface conditions, but is generally limited to the top (~1 cm depth) soil layer using moderate frequency (e.g. C-, X-band)  $T_b$  retrievals from AMSR-E and AMSR2 sensors.

The goals of this study were to (a) provide an enhanced data record over prior (v1) LPDR releases in terms of both retrieval accuracy and temporal coverage; (b) generate consistent retrievals from AMSR-E and AMSR2 suitable for long-term evaluations of key land parameters important to ecosystem processes; and (c) facilitate LPDR utility for the Earth

Science community by providing detailed descriptions of algorithm structure, retrieval accuracy and product performance, and data format specifications. The LPDR methods, data processing, global performance and uncertainty assessments are presented below.

#### 2. Methods

#### 2.1 LPDR v1 Algorithm and refinements

- In the LPDR v1 algorithms, the satellite observed microwave emission from land overlying a non-scattering atmosphere is theoretically described by three components representing the upward emission of the atmosphere, land surface upward emission attenuated by the atmosphere, and the downward atmosphere emission reflected by the land surface and attenuated by atmosphere (Wang and Manning, 2003; Jones et al., 2010). Atmosphere effects are mainly determined by air temperature, and the optical depth of oxygen, cloud liquid water, and atmosphere water vapor (Wentz and Meissner, 2000; Jones et al.,
- 2010). The land surface upward microwave emission is represented as the overall emission from a mix of land surface features including open water, vegetation and soil (Mo et al., 1982; Jones et al., 2010). Based on the above theory, the LPDR v1 algorithms derive land surface parameters in two steps: first, effective surface temperature (*Ts*),  $T_{mx}$  and  $T_{mn}$ , *fw* and *PWV* are obtained using an iterative algorithm approach that incorporates H- and V-polarized 18.7 GHz and 23.8 GHz  $T_b$  data, and several temperature insensitive microwave indices (Jones et al., 2010). In this step, a simplified land emission model that
- considers constant dry soil emissivity is adopted for facilitating the inversion process. The X-band *VOD* is then obtained by inverting the land-water microwave emissivity slope index, and surface (~ 0-1 cm depth) *mv* is acquired after correcting for X-band atmosphere, *fw* and vegetation effects (Jones et al., 2010). More detailed descriptions of the LPDR v1 algorithms

Serience Science Scien

5

20

are provided elsewhere (Jones et al., 2010). Recent refinements based on the LPDR v1 algorithm framework were carried out separately using AMSR-E or AMSR2  $T_b$  observations, including: (a) an empirical calibration of the AMSR2 *PWV* retrieval based on similar observations from AIRS (Du et al., 2015); (b) a refined AMSR2 estimation of  $T_{mx}$  and  $T_{mn}$  that considers terrain and latitude effects (Du et al., 2015); (c) an improved AMSR-E *vsm* retrieval using a weighted averaging strategy and dynamic selection of vegetation scattering albedos (Du et al., 2016a).

#### 2.2 LPDR retrieval algorithms

The latest (v2) LPDR algorithms were developed based on the available algorithm framework and improvements (Section 2.1). For generating a consistent LPDR product, the available algorithm refinements were adapted for both AMSR-E and AMSR2 portions of the combined, calibrated  $T_b$  record (Section 3.1). The final regression formulas for estimating *PWV* are

10 described below, which follow from (Du et al., 2015) but use different regression coefficients; for the satellite ascending (PM) overpass, the empirical calibration resulted in:

$$PWV_{PM} = -4.06 + 0.22T_s + \frac{A_{vd}}{a_{v23} - a_{v18}} (0.47 + 0.26\exp(-H)) - 1.63\log(\frac{\Delta T_b(89.0)}{\Delta T_b(36.0)})$$
(1)

and for the descending (AM) overpass:

$$PWV_{AM} = 1.06 + 0.27T_s + \frac{A_{vd}}{a_{v23} - a_{v18}} (0.48 + 0.21\exp(-H)) - 1.63\log(\frac{\Delta T_b(89.0)}{\Delta T_b(36.0)})$$
(2)

The *PWV* estimate is derived by a weighted sum of  $T_s$  (°C), atmosphere optical depth  $A_{vd}$  estimated from the 23.8 GHz and 18.7 GHz  $T_b$  polarization difference ratios, a cloud correction term  $\Delta T_b$  (89.0) and surface elevation H (km). The terms  $a_{v18}$  and  $\Delta T_b$  (36.0)

 $a_{v23}$  are empirically derived water vapor absorption coefficients (Jones et al., 2010). The regression formulas for estimating  $T_{mv}$  and  $T_{mn}$  are given as:

$$T_{mn} = 3.55 + 0.69T_s + 11.86T_c - 6.67T_c^2 - 0.14(abs(Lat)) + 2.74\gamma\cos(t) + 1.83*\log(fw+1.0)$$
(3)  
$$T_{mx} = 7.49 + 0.79T_s - 5.71T_c + 11.45T_c^2 - 0.14(abs(Lat)) + 2.20\gamma\cos(t) + 1.75*\log(fw+1.0)$$
(4)

where  $T_s$  is the effective surface temperature,  $T_c$  is the frequency dependent vegetation transmissivity, which is  $T_c = \exp(-VOD)$ ;  $t = 2\pi\omega - \pi$ ;  $\omega = \frac{doy}{n}$ ;  $\gamma = sign(Lat)(1 - \frac{abs(abs(Lat) - 45)}{45})$ ; doy is the day of year; *n* is the total days in a year, and

25 *Lat* is the geographic latitude; *fw* is the fractional proportion (%) of standing water cover within a grid cell, and is used for minimizing open water impacts on the temperature retrievals.

Besides the above updates, we performed additional *fw* calibration for improving the *vsm* retrievals in this study. As described above, the iterative retrieval algorithm proposed in (Jones et al., 2010) and revised in (Du et al., 2015) assumes dry soil conditions for estimating *fw*, *VOD* and atmosphere properties. Consequently, the *fw* retrieval is likely to be affected by a

30 soil moisture signal when the simplified dry soil assumption is not fully satisfied. Therefore, an empirical calibration of AMSR-E/2 *fw* was made for the purpose of improving the soil moisture inversion as follows: (a) AMSR-E *fw* values for the non-frozen period over the 2003-2010 record were averaged for each 25-km grid cell and compared with an ancillary

Searth System Discussion Science Solutions Data

MODIS 250-m MOD44W static fw map (Carroll et al., 2009); (b) the resulting AMSR-E fw summer average values were grouped into 1000 population ranges from 0.0 to 1.0 and 0.001 intervals; (c) for each group, mean AMSR-E fw and corresponding MOD44W values were calculated; and (d) relationships between AMSR-E and MOD44W fw retrievals were analyzed based on their mean group values and derived for two respective conditions: AMSR-E fw 

temperature measurements from 142 World Meteorological Organization (WMO) sites for selected years 2010 (representing AMSR-E) and 2013 (representing AMSR2). The LPDR derived *PWV* results were analyzed against AIRS *PWV* observations from the same 142 WMO site locations for the 2010 and 2013 periods. Finally, the LPDR derived daily *vsm* results were compared against independent surface soil moisture measurements from four regional soil station networks. The metrics

used to evaluate agreement between the LPDR results and independent observations included correlation coefficient (R), root

mean square error (RMSE) and bias. For evaluating LPDR consistency, only grid cells with high-quality retrievals were considered in the analysis, which excluded areas with higher vegetation biomass cover (VOD > 2.3 representing over 90% loss of underlying soil/open water signals from vegetation attenuation) or where the difference between V-pol and H-pol  $T_b$  retrievals at 18 GHz or 23 GHz was

- less than 1.0 K (i.e. indicating microwave signal saturation); grid cells containing large water bodies (fw > 0.2) were also excluded to avoid excessive contamination of the land signal by open water (Du et al., 2015; Jones, 2016). Moreover, we divided 365 (366 for leap year) days of a year into 122 three-day periods; and for each three-day period selected for the consistency evaluation, we required at least one high-quality retrieval within the period for each year of 2003-2010 and 2013-2015 portions of record. Based on the above data selection criteria, the global monthly mean of the high-quality LPDR
- daily estimates were calculated for each month of the AMSR-E (2003-2010) and AMSR2 (2013-2015) full-year records and analyzed using statistical metrics including mean, SD and range.

#### 3. Data processing and ancillary datasets

#### 3.1. AMSR-E and AMSR2 T<sub>b</sub> records used for land parameter retrievals

Multi-frequency  $T_b$  observations from AMSR-E and AMSR2 provide the primary inputs for LPDR processing. The AMSR-20 E sensor was launched on 4 May 2002 onboard the NASA EOS Aqua satellite and operated until 4 October 2011. AMSR-E was succeeded by AMSR2, which was launched on 18 May 2012 on-board the JAXA GCOM-W1 satellite. Both sensors provide global measurements of vertically (V) and horizontally (H) polarized microwave emissions at six frequencies (6.9, 10.7, 18.7, 23.8, 36.5, 89.0 GHz) with descending/ascending orbital equatorial crossings at 1:30 AM/PM local time. Though succeeding most characteristics of its predecessor, AMSR2 is different from AMSR-E in several aspects including (a) an

- additional frequency at 7.3 GHz designed for mitigating Radio Frequency Interference (RFI); (b) a larger (2.0 m diameter) main reflector providing enhanced spatial resolution retrievals, and (c) an improved calibration system (Imaoka et al., 2010). For developing a consistent global land parameter record, the AMSR-E/2  $T_b$  retrievals were pre-processed in four steps: (a) AMSR-E orbital swath  $T_b$  data from the Remote Sensing Systems (RSS) Version 7 product were spatially re-sampled and reprojected to a 25-km resolution global Equal Area Scalable Earth (EASE) Grid Version 1 format following previously
- 30 established methods (Armstrong and Brodzik, 1995; Ashcroft and Wentz, 1999; Brodzik and Knowles, 2002). In this study, an additional altitude correction of the  $T_b$  orbital swath retrievals was made to improve sensor footprint geolocation accuracy prior to the gridding process. The altitude correction to the AMSR2 L1R data considers the actual surface of the Earth

instead of an ideal Earth ellipsoid (Maeda et al., 2016), which helps to ensure reliable analysis of AMSR-E/2 land surface retrievals over high elevation areas, including the Qinghai-Tibetan Plateau; (b) a similar gridding process was performed on the AMSR2 L1R swath data; (c) the AMSR2 multi-frequency (X- to W-band) *T<sub>b</sub>* retrievals were empirically calibrated against the same AMSR-E channels using similar overlapping *T<sub>b</sub>* observations from the Microwave Radiation Imager
(MWRI) on-board the Chinese FY3B satellite (Du et al., 2014). However, in contrast with Du et al. (2014) where the *T<sub>b</sub>* calibration was conducted on per grid cell basis for each frequency, polarization and orbit, the approach used for this investigation involved calibrating within 5×5 grid cell windows to minimize the impact of the different sensor footprints. Both ascending and descending orbit X-band *T<sub>b</sub>* data for a given polarization were calibrated together because the largest differences and lowest correlations were found between overlapping MWRI and AMSR-E/2 X-band observations among all sensor frequencies utilized (Du et al., 2014); the combined orbit X-band *VOD* retrievals, which are particularly sensitive to *T<sub>b</sub>* calibration uncertainties, especially for higher vegetation biomass conditions; (d) finally, the gridded and calibrated AMSR-

effects from RFI, active precipitation, frozen conditions, and permanent ice and snow cover using previously established 15 methods (Jones et al., 2010). The  $T_b$  screening under frozen land surface conditions was identified using an existing global daily freeze-thaw (FT) data record derived from a refined classification algorithm (Kim et al., 2017) and AMSR-E/2 36.5 GHz V polarized  $T_b$  retrievals in a consistent 25-km resolution global EASE-grid projection format; the FT mask is represented as a grid cell-wise daily binary bit flag in the LPDR data set and was used to identify and screen frozen land surface conditions from further LPDR processing and retrievals.

 $E/2 T_b$  data were subjected to additional screening prior to implementing the retrieval algorithms to minimize potential noise

#### 20 **3.2.** Ancillary data used for algorithm calibration and LPDR performance assessment

A variety of ancillary data were used for calibrating the LPDR algorithms and evaluating LPDR global performance. The ancillary data included atmosphere *PWV* retrievals from AIRS (Divakarla et al., 2006), a static MOD44W open water map (Carroll et al., 2009), GIMMS3g NDVI (Pinzon and Tucker, 2014) and in situ surface soil moisture measurements from four globally distributed measurement networks (Jackson et al., 2010; Yang et al., 2013; Smith et al., 2012). All ancillary data

25 were re-projected to the same 25 km EASE-grid Version 1 format as the LPDR to facilitate algorithm calibration and product comparisons.

The AIRS *PWV* products were used for LPDR *PWV* algorithm calibration and product comparisons. The LPDR iterative retrieval algorithm for *PWV* (Jones et al., 2010; Section 2.1) was empirically calibrated and quantitatively validated using synergistic *PWV* observations (version 6 level 2 swath product) from AIRS and the Advanced Microwave Sounding Unit

30 (AMSU) instruments (Du et al., 2015). Both AIRS and AMSU are deployed on the Aqua satellite together with AMSR-E and have the same local overpass time as AMSR2. The AIRS Version 6 product is expected to have higher accuracy than the previous AIRS Version 4 water vapor record, which shows retrieval uncertainties less than 15% in comparison with radiosonde measurements in 2-km troposphere layers (Divakarla et al., 2006; Diao et al., 2013).

For calibrating LPDR derived *PWV*,  $T_{mx}$  and  $T_{mn}$  retrievals over different land cover types, in-situ daily  $T_{mx}$  and  $T_{mn}$  measurements were obtained along with coincident AIRS *PWV* retrievals for year 2010 from 250 globally distributed WMO weather station locations (Fig.1). The spatial distribution of WMO stations selected was designed to be representative of major global land cover classes (Justice et al., 2002; Friedl et al., 2010). The WMO air temperature record was obtained from the National Climate Data Center (NCDC) Global Summary of the Day (GSOD version 7) using previously established criteria (Jones et al., 2010). The calibration was made for year 2010 and the derived relationships were applied to the entire AMSR-E/2 record. Independent daily air temperature measurements and collocated AIRS *PWV* retrievals from 142 other globally distributed WMO weather stations (Fig.1) operating from years 2010 to 2013 were selected for evaluation of LPDR derived  $T_{mx}$ ,  $T_{mn}$  and *PWV* accuracy; relative consistency in performance between AMSR-E (represented by year 2010) and AMSR2 (represented by year 2013) portions of the LPDR record was also assessed.

The LPDR derived *fw* record was evaluated against the higher-resolution (250-m), global-scale MOD44W static open water product (Carroll et al., 2009). The MOD44W product is derived from a compilation of the SRTM (Shuttle Radar Topography Mission) Water Body dataset and the MODIS MOD44C Collection 5 (2000–2008) open water classification (Haran, 2005; Carroll et al., 2009). The MOD44W map was re-projected and aggregated to the same 25 km EASE-grid format as the LPDR prior to the comparisons

format as the LPDR prior to the comparisons.

The LPDR derived *VOD* record was evaluated over the global domain using synergistic satellite optical-IR observations of vegetation greenness defined from NDVI. The GIMMS3g (version 1) global NDVI record derived from calibrated NOAA Advanced Very High Resolution Radiometer (AVHRR) sensor observations (Pinzon and Tucker, 2014) has been widely used in evaluating global vegetation status and changes (Zhu et al., 2016); the bi-monthly NDVI data were re-projected from

- their native 1/12 degree spatial resolution and geographic projection format to the same 25-km resolution global EASE-grid format as the LPDR for the 2003 to 2015 record. The NDVI is sensitive to changes in vegetation greenness and differs from LPDR derived 10.65 GHz VOD sensitivity to canopy biomass and water content variations, including both photosynthetic (e.g. foliar) and non-photosynthetic (e.g. stem and branch) elements (Jones et al. 2013). Both satellite NDVI and VOD records have been shown to provide similar synergistic canopy phenology metrics distinguishing both seasonal and spatial
- differences among different plant functional types (Jones et al. 2011).

The LPDR *vsm* retrieval accuracy was evaluated using a similar approach as (Du et al., 2016) by comparing the satellite X-band (10.65 GHz) daily soil moisture retrievals against collocated in situ surface soil moisture measurements from four globally distributed soil moisture measurement networks (Fig.1). The Little River network (LR; centroid 83.61 °W, 31.65 °N) has a humid climate representing forest, cropland and pasture vegetation (Jackson et al., 2010). The Little Washita network

(LW; centroid 98.1°W, 34.95°N) has a sub-humid climate dominated by rangeland and pasture vegetation (Jackson et al., 2010). A three-year (2003 to 2005) LR and LW daily soil moisture record representing surface (0-5 cm depth) soil layer conditions was used for this study. The Naqu (NQ; centroid 91.875°E, 31.625°N) soil moisture network was located on the Tibetan Plateau in western China. Surface soil moisture measurements extending from August 2010 to September 2011 from the NQ network were used for evaluating LPDR performance in an environment characterized as high elevation, with large

surface soil moisture variability and sparse vegetation (Chen et al., 2013; Yang et al., 2013). The Yanco (YC; centroid 146.0915°E, 34.842°S) network is part of the larger Murrumbidgee Soil Moisture Monitoring Network (MSMMN) in Australia (Smith et al., 2012; Panciera et al., 2014) and represents a Southern Hemisphere semi-arid agricultural and grazing landscape; a two-year (2009-2010) YC surface soil moisture record was also used for this study.

#### 5 4. Results

#### 4.1 Fractional open water

The LPDR fw composites (Fig. 2a) for non-frozen periods capture characteristic global inundation patterns consistent with the ancillary MOD44W static water map (Fig. S1), including extensive wetland complexes in the pan-Arctic region, Bangladesh and Argentina, and major river systems such as the Amazon, Mississippi, Yangtze, and Yenisei. Large fw

- 10 seasonal variations (Fig. 2b) associated with seasonal precipitation and/or snow melt events occur over the Mississippi basin, Parana River Basin, northern Canada and Eurasia, Indian sub-continent, southern Tibet, and eastern China. The LPDR *fw* record also distinguishes dynamic flooding events not represented by the ancillary static water map, including extensive water inundation (Fig.2a) and large seasonal *fw* variations (Fig.2b) in Bangladesh where the summer monsoon brings large precipitation driven flooding (Brouwer et al., 2007).
- Quantitative comparisons of LPDR *fw* annual means in relation to MOD44W were made for respective AMSR-E (2003-2010) and AMSR2 (2013-2015) full-year records (Table 1). Both AMSR2 and AMSR-E *fw* annual means show favorable spatial correspondence with the MOD44W results ( $R \ge 0.75$ ,  $RMSE \le 0.06$ ). The LPDR inundated area percentage also shows a mean 1.50% wet bias relative to the MOD44W product, which may partially result from better LPDR microwave sensitivity to surface water dynamics, including water beneath vegetation (Du et al., 2016b). Higher LPDR *fw* levels along
- 20 coastlines are due to larger water cover of coastal grid cells within the land mask. The LPDR results also show generally larger coastal *fw* levels than MOD44W, indicating ocean contamination of adjacent land grid cells within the coarser AMSR- $E/2 T_b$  footprint.

#### 4.2 Atmosphere precipitable water vapor

The spatial distributions of LPDR *PWV* climatology mean (Fig.3a) and SD (Fig.3b) results derived from ascending orbit  $T_b$  retrievals and full-year observations were compared with benchmark satellite *PWV* retrievals from AIRS (Fig. S2). Both LPDR and AIRS *PWV* retrievals show similar global patterns and latitudinal distributions, with generally higher water vapor levels at lower latitudes and warmer climate zones, consistent with the near-exponential relationship between atmospheric temperature and moisture holding capacity, except for dry desert regions distinguished by lower characteristic *PWV* levels. Especially large *PWV* levels are observed over the Bay of Bengal and adjacent regions (Fig.3a) where a large amount of

30 water vapor is transported by the summer monsoon (Uma et al., 2014). Large PWV seasonal variations (SD) are apparent in

regions with distinct dry and wet seasons, including the Indian sub-continent, eastern China and the African Sahel (Fig.3b); these spatial and temporal patterns are consistent between the LPDR and AIRS products. The LPDR shows larger PWV seasonal variability in tropical rainforest regions (Fig.3b) than the AIRS observations, which is attributed to ill-conditioned LPDR retrieval associated with microwave signal saturation over dense vegetation cover. Overall, the LPDR and AIRS

5 ascending and descending orbit derived PWV monthly means are highly correlated (R = 0.99) (Fig.4) with a major peak in the Northern Hemisphere summer months (July and August) and a secondary peak in the Southern Hemisphere summer months (January and February).

The LPDR PWV retrievals were quantitatively validated against the AIRS observations at 142 global WMO weather station locations for years 2010 and 2013 (Table 2). The AMSR-E/2 retrievals show strong agreement with the AIRS PWV product ( $R \ge 0.91$ ; RMSE  $\le 4.99$  mm), though a slight *PWV* over estimation and under estimation are indicated for respective

AMSR-E (bias  $\leq 0.27$  mm) and AMSR2 (bias  $\geq -0.27$  mm) portions of record (Table 2).

#### 4.3 Daily maximum/minimum surface air temperature

The LPDR derived global mean and SD variability maps for  $T_{mx}$  are presented in Figure 5, while the  $T_{mn}$  results show similar global and seasonal patterns. The LPDR results show characteristic global temperature patterns following major climate 15 zones and latitudinal gradients, and similar to the PWV results (Fig. 3), but with generally greater surface spatial complexity influenced by proximity to coastal areas, vegetation and land cover conditions, and elevation-driven temperature lapse rates (Du et al., 2015). The LPDR results show expected smaller seasonal temperature variability (SD) near the equator and larger variability at higher latitudes, especially in the interior of large landmasses such as North America and Asia. The resulting temperature maps (Fig. 5) only represent non-frozen land surface conditions rather than complete annual cycles (i.e. sections

20 2.3, 3.1).

10

The LPDR derived  $T_{mx}$  and  $T_{mn}$  retrievals were validated against independent in-situ daily air temperature measurements from 142 global WMO weather stations for years 2010 and 2013 (Table 2). Overall, the LPDR temperatures corresponded favorably with the WMO temperature measurements ( $R \ge 0.90$ ; RMSE  $\le 3.48$  °C). The AMSR-E (2010) and AMSR2 (2013) retrievals show similar  $T_{mx}$  and  $T_{mn}$  retrieval accuracy, with associated RMSE differences within 0.16 K in relation to the

- WMO daily temperature measurements. These results indicate improved LPDR temperature accuracy relative to previously 25 reported AMSR2 derived accuracies for  $T_{mx}$  (RMSE = 3.64 °C) and Tmn (RMSE = 3.54 °C) from a prior study (Du et al., 2014); the higher LPDR temperature accuracy (RMSE  $\leq$  3.48 °C) suggests an improvement in sensor inter-calibration and algorithm refinements (Section 3.1). However, the calibrated AMSR2  $T_b$  is not identical to that of AMSR-E as reflected by a maximum 0.38 C difference in their  $T_{mx}$  and  $T_{mn}$  retrieval biases against WMO measurements (Table 2). To evaluate the
- 30 impact of the fractional water corrections on the LPDR v2 air temperature retrievals, Eqs (1-4) were re-derived using the same procedure (section 2.2) but assuming zero fractional water cover. The results indicated approximately 13% improved RMSE performance in the  $T_{mx}$  and  $T_{mn}$  retrievals using the fw correction relative to air temperature retrievals derived without accounting for fractional water influence.

#### 4.4 Vegetation optical depth

The previous UMT LPDR v1 AMSR-E *VOD* record was assessed globally (Jones et al., 2011) and has been used for a range of regional ecosystem studies including vegetation phenology and disturbance recovery assessments (Liu et al., 2013, Jones et al. 2013, Jones et al. 2014). The *VOD* record can also be used as a data quality mask for the *vsm* retrievals because soil moisture retrieval accuracy is generally degraded under higher vegetation biomass levels (Du et al., 2016a). In this study, the LPDR derived *VOD* was compared with the GIMMS3g NDVI record based on an assumption of proportionality between vegetation canopy biomass and greenness variations (Jones et al. 2011). The LPDR *VOD* pattern and seasonal variability (SD) are generally consistent with the global pattern in vegetation cover indicated from the NDVI record (Fig. S3). The LPDR derived mean annual *VOD* results (Fig.6a) show characteristic global patterns in vegetation biomass, including higher

- VOD in tropical rainforests (e.g. Amazon Basin, Congo Basin Southeast Asia) and much lower VOD in arid and sparsely vegetated areas, including the Sahara and Sonoran deserts, and Central Australia. Moderate VOD levels occur in grassland, shrubland and cropland areas, including the Central USA, sub-Saharan Africa, central China and India. Larger VOD relative seasonal variability (Fig.6b) (i.e. VOD SD normalized by the mean; %) occurs over predominantly deciduous and lower biomass areas, including grassland, shrubland and cropland. Large VOD seasonal variations also occur in semi-arid climate
- zones with distinctive wet and dry cycles, including the African Sahel where plant growth depends on seasonal rainfall (Proud and Rasmussen, 2011). A few *VOD* change hotspots occur in wetland areas (e.g. Iber á Wetlands in Argentina and Bangweulu Wetlands in Zambia), which may reflect emergent vegetation overlying a standing water background during the wet season. Lower *VOD* seasonality occurs in the tropics, consistent with a smaller seasonal climate cycle near the equatorial zone. Arid areas show generally low *VOD* levels and seasonality consistent with sparse vegetation cover, except
- for some areas, including portions of Arabian Peninsula, where relatively large *VOD* seasonality may be a result of irrigation activities (Siebert et al., 2005).

Both *VOD* and NDVI display similar seasonal cycles represented by their mean monthly time series ( $R \ge 0.88$ ), but with temporal phase offsets in *VOD* and NDVI cycles for different land cover types (Fig.7). Here, the mean seasonal cycle in *VOD* and NDVI is depicted for major IGBP global land cover types, including evergreen needleleaf forest (ENF), evergreen

- broadleaf forest (EBF), deciduous needleleaf forest (DNF), deciduous broadleaf forest (DBF), grassland and cropland. The LPDR *VOD* and GIMMS3g NDVI comparison results are summarized in Table 3 and show strong correspondence for both ascending ( $0.67 \le R \le 0.90$ ) and descending ( $0.84 \le R \le 0.95$ ) orbit retrievals for ENF, DNF, grassland and cropland areas with relatively well-defined seasonal cycles. A *VOD* temporal phase shift relative to NDVI is evident for croplands and detectable for other land cover types, reflecting different vegetation biophysical properties that the microwave and optical-
- infrared instruments are sensitive to (Jones et al., 2011, 2012). Weaker *VOD* and NDVI correlations in EBF regions coincide with lower characteristic canopy seasonality in the tropics, but may reflect degraded signal-to-noise due to persistent cloud and atmospheric aerosol effects limiting effective NDVI retrievals, and *VOD* and NDVI saturation over dense canopies (Jones et al., 2011). The *VOD* estimates derived from the descending orbit  $T_b$  retrievals also show overall stronger correspondence with NDVI than the ascending retrievals, especially for DBF regions (descending orbit R = 0.87; ascending

orbit R = 0.20). Differences in NDVI correspondence between the ascending and descending orbit *VOD* records may reflect regional *VOD* retrieval uncertainties contributed by deficiencies in the underlying LPDR algorithm assumptions and parameterizations, which are discussed below (Section 5).

#### 4.5 Soil Moisture

- The global soil moisture pattern depicted by the LPDR X-band *vsm* record (Fig. 8) is generally consistent with the known global climatology, including characteristically wet surface soil moisture conditions in northern high latitude areas and drier soil moisture extremes in deserts, and semi-arid regions such as the African Sahara, southwest USA, and central Australia. Wetter *vsm* conditions along coastal boundaries may reflect remaining ocean  $T_b$  contamination of adjacent land grid cells within the coarser sensor footprint despite explicit *fw* correction of the *vsm* retrievals. Relatively large seasonal soil moisture
- variations are associated with areas having distinctive wet and dry seasons, including the African Sahel, central USA, Indian subcontinent and southern Tibet. For arid regions such as central Australia, high relative (%) seasonal SD variability is due to low average *vsm* conditions. Lower *vsm* variability occurs over higher vegetation biomass (*VOD*) areas, including forests, where AMSR-E/2 soil moisture sensitivity and *vsm* retrieval performance are expected to be lower due to loss of soil sensitivity; the global range of effective *vsm* retrievals and other LPDR observations are represented by the data quality 15 metries described helem (Sentier 5.2).
- metrics described below (Section 5.2).

The LPDR *vsm* retrievals were compared against globally distributed validation watershed measurements (Table 4). The LPDR results show overall favorable *vsm* accuracy in relation to independent in situ soil moisture observations from the globally distributed monitoring sites within the effective LPDR domain ( $0.63 \le R \le 0.84$ ;  $0.03 \le$  bias corrected RMSE  $\le 0.06 \text{ cm}^3/\text{cm}^3$ ). These results indicate similar or better accuracy than the reported performance of other AMSR-E soil

moisture products (Jackson et al., 2010; Du et al., 2016a), and generally better LPDR performance for descending (AM) than ascending (PM) orbit *vsm* retrievals.

#### 5. Discussion

The latest (v2) LPDR incorporates recent algorithm refinements and updates over the original baseline algorithms and data record (Jones et al., 2010), while also including an extended global data record spanning both AMSR-E and AMSR2

- observation periods (Jun. 2002 Dec. 2015). The resulting data record produces global environmental patterns and seasonal dynamics consistent with characteristic climate and land cover variability; the LPDR also shows favorable agreement with a diverse set of independent observation benchmarks. The LPDR algorithms and parameter estimates are internally consistent and include an integrated set of environmental parameters representing atmosphere, vegetation, surface and soil conditions derived from simultaneous satellite multi-frequency  $T_b$  observations. The iterative algorithm and multi-parameter retrieval
- approach enable decomposition of the satellite observations into atmosphere, vegetation, standing water and soil moisture

components. In particular, the dynamic open-water (*fw*) correction in the LPDR algorithm benefits *vsm* retrievals over areas with significant spatial and seasonal inundation variability.

#### 5.1 LPDR data format

The resulting LPDR is available in a 25-km resolution global EASE-Grid (v1) projection and GeoTIFF file format. The data

- 5 files are organized by ascending and descending orbits on a daily basis. Each GeoTIFF file consists of six 2-D (1383 columns, 586 rows) data arrays representing five float-type retrieval data bands (*fw*,  $T_{mx}$  or  $T_{mn}$ ,  $T_c$ , *PWV*, *vsm*) and one byte-type QC band. A set of product QC flags are included to inform the user about the estimated quality of retrieved parameters or missing data. The QC binary bit flags are summarized in Table 5 and indicate the presence or absence of the following land surface conditions: frozen ground (1<sup>st</sup> bit), snow or ice presence (2<sup>nd</sup> bit), strong precipitation (3<sup>rd</sup> bit), RFI at
- 10 18.7 GHz (4<sup>th</sup> bit), RFI at 10.65 GHz (5<sup>th</sup> bit), dense vegetation with VOD > 2.3 (6<sup>th</sup> bit) large water bodies with fw > 0.2 (7<sup>th</sup> bit), and saturated microwave signals (difference between V-pol and H-pol brightness temperature at 18 GHz or 23 GHz less than 1.0 K) (8<sup>th</sup> bit).

#### 5.2. Data record consistency

The LPDR record described in this study extends from Jun. 2002 to Dec. 2015 and captures both short-term (diurnal, daily, annual) variability and longer-term (annual, decadal) climate trends over the global terrestrial environment for a diverse set of significant environmental parameters. Potential differences in  $T_b$  characteristics and algorithm performance between AMSR-E and AMSR2 portions of the LPDR are expected to introduce artifacts and degrade LPDR precision for analyzing long-term environmental changes. LPDR data consistency was examined through statistical comparison of best quality (QC) retrievals between AMSR-E and AMSR2 portions of record (Section