# Peer review of "A global satellite environmental data record derived from AMSR-E and AMSR2 microwave earth observations"

_Earth System Science Data, 2017_

## Editor Comment (EC1) · M. E. Contadakis (Editor) · 21 Apr 2017

Dear readers . The last days appears in the ESSDD Journal a remarkable Review article A global satellite environmental data record derived from AMSR-E and AMSR2 microwave earth observations. by Jinyang Du et al.. As the responsible for the handling of the article Editor I encourage you to contribute to the discussion by submitting any comment or question or remark, which according to your opinion, would ameliorates the article.

---

## Referee Comment (RC1) · Anonymous Referee #1 · 16 May 2017

This ms describes a valuable data product for high temporal resolution land surface dynamics (Land Parameter Data Record; LPDR) constructed from two passive microwave time series (AMSR-E and AMSR2). The variables retrieved include surficial volumetric soil moisture, air temperature at 2 m, precipitable water vapor, fractional open water, and vegetation optical depth (VOD), all retrieved at 25 km resolution for ascending and descending passes.

I have just a few concerns about this well-written ms.

The authors should reconsider labeling the surface air temperature retrievals as the daily Tmax and Tmin. There is no guarantee that these are the diel extrema common to meteorological and climatological usage since there are a limited number observations per day over much of the land surface. These observations are very valuable particularly in those many parts of the planet where there are few temperature stations or the accessibility to the temperature data is restricted.

The authors have chosen to display the dispersion about mean as standard deviation in units in some maps but as "temporal SD variability (%)" in others. Displaying the temporal variation consistently the percent coefficient of variation (%CV=100*SD/mean) would make it easier to understand relative variation of each retrieved quantity across continents, biomes, and ecoregions.

The authors choose to use the AVHRR GIMMS-3g NDVI dataset to compare with the LPDR VOD data. Why? The MODIS data record (for instance, NBAR at CMG resolution) offers a higher SNR.

Can we see a table of the land area with high QC retrievals by season?

Are the Spearman correlations substantially different from the Pearson correlations?

Can you provide a third digit for the RMSD and bias values in Tables 1, 2, and 4?

How about a specific comment on the odd descending correlation between VOD and NDVI for EBF?

The figures need tuning to improve their accessibility. About 8% of the male population suffers from some degree of difficulty distinguishing red from green (aka color blindness). Figures 1-3, 5-6, 8, and S1-S3 all use red and green together. Consult http://colorbrewer2.org for better colormaps. Figure 1 is too busy. Consider reducing the number of cover classes which should help the color issue. The blue circles are particularly difficult to find against the background.

Perhaps I missed them at the distribution portal, but where are the geospatial metadata for the LPDR that are compliant with either FGDC or ISO standards?

---

## Referee Comment (RC2) · Anonymous Referee #2 · 19 Aug 2017

General comments

Thank the authors for this important work! AMSR-E/2 has long been used to support global eco-hydrological studies. Various land/atmosphere states have been developed independently based on a single or multi-channels of AMSR-E/2 brightness temperature observations. This work, however, has integrated previous studies, aiming to provide an "internally consistent" environmental dataset based on AMSR-E/2. Started with some background introduction, this manuscript documents the LPDR retrieval algorithms as well as the refinements, followed by a comprehensive global evaluation and some discussions on the limitations and uncertainties of this integrated dataset.

The topic of this work definitely fits ESSD, and I recommend this manuscript being published on ESSD by considering the following comments/suggestions.

Specific comments

1. Snow (e.g., snow water equivalent) and freeze-thaw products are also common retrievals from AMSR-E/2 observations, and I am wondering why they are excluded in the current dataset. I realize it is difficult to include all the retrievals at a time but I still expect a short discussion from the authors regarding this issue.

2. The word "internal consistency", which is one of the most important motivations of this work, appears many times throughout the manuscript. However, it is still unclear to me how is this "consistency" preserved or reflected in the dataset. To me, all retrievals from one single sensor does not guarantee "consistency" as they may be obtained by using distinct retrieval algorithms. I therefore expect the authors to reorganize sections 2.1-2.2 and to more explicitly explain how these five retrievals are "internally" connected. To this end, a diagram or flowchart showing the general retrieval process and their physical connections is highly desired.

3. For soil moisture evaluation, there are representativeness issues – both horizontally and vertically. For the former, how do you upscale the point-scale soil moisture measurements into a 25km grid-scale? A map showing the spatial distributions of soil moisture stations and the corresponding AMSR-E grid-cells should also help. For the latter, as mentioned in the manuscript, most of the in-situ surface soil moisture observations are recorded at 5 cm or 0-5 cm depth, while AMSR-E retrieval only represents wetness conditions within the top 1 cm. These two issues can potentially introduce "biased" evaluation on vsm, please clarify.

4. P5, L1-5: is the "empirical calibration" kind of CDF-matching AMSR-E/2 to the climatology of MODIS fw? Meanwhile, at P5, L5: how is the threshold of fw=0.15 determined?

Technical corrections

1. P4, L10: "(Du et al. 2015)" should be "Du et al. (2015)". Also see citations at P4, L28, and P8, L26.

2. P10, L26: in "Tmn", "mn" should be subscript.
* * *

---

## Author Comment (AC1) · 11 Sep 2017

Dear Editor:

Thank you for your comments on our work. Please find a detailed point-by-point response to the review comments with changes in the revised manuscript highlighted.

Best regards,

Jinyang Du, John S. Kimball, Lucas A. Jones, Youngwook Kim, Joseph Glassy, and Jennifer D. Watts

---

## Author Comment (AC2) · 11 Sep 2017

Please check the attached file.

Please also note the supplement to this comment:
https://www.earth-syst-sci-data-discuss.net/essd-2017-27/essd-2017-27-AC2-supplement.pdf

———————————————————

---

## Author Comment (AC3) · 11 Sep 2017

**Reply to RC2 for the manuscript "A global satellite environmental data record derived from AMSR-E and AMSR2 microwave earth observations" by Du Jinyang, John S. Kimball, Lucas A. Jones, Youngwook Kim, Joseph Glassy, and Jennifer D. Watts.**

Dear Anonymous Referee #2, thank you for your constructive comments on our manuscript. Please find below our responses to all the comments (in **_bold and italic_**). The changes on the manuscript were highlighted in blue.

*General comments*
***Thank the authors for this important work! AMSR-E/2 has long been used to support global eco-hydrological studies. Various land/atmosphere states have been developed independently based on a single or multi-channels of AMSR-E/2 brightness temperature observations. This work, however, has integrated previous studies, aiming to provide an "internally consistent" environmental dataset based on AMSR-E/2. Started with some background introduction, this manuscript documents the LPDR retrieval algorithms as well as the refinements, followed by a comprehensive global evaluation and some discussions on the limitations and uncertainties of this integrated dataset.***
***The topic of this work definitely fits ESSD, and I recommend this manuscript being published on ESSD by considering the following comments/suggestions.***

Reply: Thank you for the comments, summary and recommendations; we attempted to address all of the reviewer comments and recommendations in the revised manuscript as summarized below.

*Specific comments*
***Snow (e.g., snow water equivalent) and freeze-thaw products are also common retrievals from AMSR-E/2 observations, and I am wondering why they are excluded in the current dataset. I realize it is difficult to include all the retrievals at a time but I still expect a short discussion from the authors regarding this issue.***

Reply: Thank you for the suggestion. In a separate study, we developed an independent freeze-thaw product based on 36 GHz $T_b$ observations of AMSR-E/2 as also mentioned in the revised manuscript(Page 7; Line 27-28) "The $T_b$ screening under frozen land surface conditions was identified using an existing global daily freeze-thaw (FT) data record derived from a refined classification algorithm (Kim et al., 2017)". The freeze-thaw record is also represented in the LPDR as a simple daily frozen flag, which was used for screening out frozen conditions prior to deriving the other land parameter retrievals (Figure 1). It would be interesting and very useful to extend the LPDR framework to include snow properties and frozen soil conditions, though the microwave emission and scattering from snow, ice, frozen vegetation and soil need to be carefully modeled. As suggested by the reviewer, we added a short discussion in the Section 5 "Discussion" as follows:

"The iterative algorithm and multi-parameter retrieval approach enable decomposition of the satellite observations into atmosphere, vegetation, standing water and soil moisture components.

In particular, the dynamic open-water (*fw*) correction in the LPDR algorithm benefits *vsm* retrievals over areas with significant spatial and seasonal inundation variability. The current algorithm is limited to non-frozen land surface conditions determined using an independent AMSR-E/2 FT product (Kim et al., 2017), while the FT parameter is represented as simplified daily frozen flag in the LPDR. Potential extension of the LPDR to represent snow cover properties and frozen conditions would enable continuous land parameter observations over a full annual cycle, while incorporating observations of other key environmental indicators of the changing cryosphere. The complex microwave emission and scattering signatures of snow, lake ice, frozen soil and vegetation must first be carefully accounted for to enable further development and extension of the LPDR retrieval algorithms (Tedesco et al., 2010; Du et al., 2017)."

The added references are listed below:

"Tedesco, M. and Narvekar, P.S.: Assessment of the NASA AMSR-E SWE product. IEEE J. Sel. Topics Appl. Earth Observ. in Remote Sens., 3(1), 141-159, 2010.

Du, J., Kimball, J.S., Duguay, C., Kim, Y. and Watts, J.D.: Satellite microwave assessment of Northern Hemisphere lake ice phenology from 2002 to 2015, The Cryosphere, 11, 47-63, 2017."

***2. The word "internal consistency", which is one of the most important motivations of this work, appears many times throughout the manuscript. However, it is still unclear to me how is this "consistency" preserved or reflected in the dataset. To me, all retrievals from one single sensor does not guarantee "consistency" as they may be obtained by using distinct retrieval algorithms. I therefore expect the authors to reorganize sections 2.1-2.2 and to more explicitly explain how these five retrievals are "internally" connected. To this end, a diagram or flowchart showing the general retrieval process and their physical connections is highly desired.***

Reply: We accepted the reviewer's suggestion. Sections 2.1-2.2 were revised for a better description of the algorithm with an additional Figure (Figure 1 of the revised manuscript) plotted to illustrate the retrieval process.

The following sentences were added to Section 2.1 as follows:
"The AMSR-E/2 frequencies have variable sensitivity to land and atmosphere properties, and the frequency-dependent optical depth of vegetation or atmospheric layers determines the degree to which measured microwave emissions originate from the soil, vegetation or atmosphere (Jones et al., 2016). The C- and X-band AMSR-E/2 measurements are generally used for inferring soil moisture under vegetation and atmosphere layers while higher $T_b$ frequencies (> 18 GHz) show relatively greater sensitivity to atmospheric properties (Njoku et al., 2003). In addition, open water may significantly impact the measured microwave emissions at all AMSR-E/2 frequencies due to the high dielectric constant of water (Jones et al., 2010; Du et al., 2016b). Based on the above theory and considerations, the LPDR v1 algorithms utilize observations at relatively high frequencies (> 18 GHz) to estimate *PWV* and *fw*; and then apply the inferred information to derive the X-band *VOD* and *vsm* retrievals. The two-step retrieval process is detailed as follows: first, effective surface temperature (*Ts*), $T_{mx}$ and $T_{mn}$, *fw* and *PWV* are obtained using an iterative

algorithm approach that incorporates H- and V-polarized 18.7 GHz and 23.8 GHz $T_b$ data, and several temperature insensitive microwave indices (Jones et al., 2010).".

The following sentence and Figure 1 were added to the end of Section 2.2:
"The general LPDR retrieval process is illustrated in Figure 1."

[Figure]

**Figure 1: The LPDR algorithm retrieval process.**

*3. For soil moisture evaluation, there are representativeness issues – both horizontally and vertically. For the former, how do you upscale the point-scale soil moisture measurements into a 25km grid-scale? A map showing the spatial distributions of soil moisture stations and the corresponding AMSR-E grid-cells should also help. For the latter, as mentioned in the manuscript, most of the in-situ surface soil moisture observations are recorded at 5 cm or 0-5*

*cm depth, while AMSR-E retrieval only represents wetness conditions within the top 1 cm. These two issues can potentially introduce "biased" evaluation on vsm, please clarify.*

Reply: We agree with the reviewer. The horizontal and vertical representativeness of in-situ soil moisture measurements is a key issue in evaluating satellite soil moisture products.

In the efforts to mitigate the spatial scale disparity between in-situ and satellite measurements, averaged soil moisture over instrumented regional networks that approximate the size of the satellite footprints has been adopted for validating satellite products (Jackson et al., 2010). The four soil moisture networks used for this study were designed for validating satellite retrievals at several 10s kilometers and their detailed descriptions are presented in Jackson et al. (2010), Smith et al. (2012), and Yang et al. (2013). To highlight the issue, the following sentences were added in Section 3.2:

"The LPDR *vsm* retrieval accuracy was evaluated using a similar approach as (Du et al., 2016) by comparing the satellite X-band (10.65 GHz) daily soil moisture retrievals against collocated in situ surface soil moisture measurements from four globally distributed soil moisture measurement networks (Fig.1). The four soil moisture regional networks represent the approximate spatial heterogeneity and sensing depth as the AMSR-E/2 $T_b$ footprint retrievals and were designed for validating satellite regional soil moisture retrievals as detailed in Jackson et al. (2010), Smith et al. (2012) and Yang et al. (2013)."

For the vertical representativeness, it is ideal to have ~0-1 cm soil moisture measurements as the reviewer pointed out; however, major soil moisture records available for regional and long-term validations are the measurements for soil layers at 0- to 5-cm or deeper. We added the following sentences to highlight the possible impacts of inconsistency in horizontal and vertical representativeness between in-situ and satellite measurements in section 4.5:

"The LPDR *vsm* retrievals were compared against globally distributed validation watershed measurements (Table 4). The LPDR results show overall favorable *vsm* accuracy in relation to independent in situ soil moisture observations from the globally distributed monitoring sites within the effective LPDR domain ($0.63 \leq R \leq 0.84$; $0.03 \leq$ bias corrected RMSE $\leq 0.06$ cm3/cm3). The apparent retrieval biases (-0.10 to 0.09) may partially reflect inconsistencies in horizontal and vertical representativeness between the in-situ soil moisture measurements and AMSR-E/2 $T_b$ retrievals (Du et al., 2016)."

**4. P5, L1-5: is the "empirical calibration" kind of CDF-matching AMSR-E/2 to the climatology of MODIS fw? Meanwhile, at P5, L5: how is the threshold of fw=0.15 determined?**

Reply: As stated in the revised manuscript (Page 5; Line 11-14), the resulting AMSR-E *fw* values were first grouped into 1000 population ranges from 0.0 to 1.0 and 0.001 intervals; and the relationships between AMSR-E and MOD44W *fw* retrievals were analyzed based on the group mean values of AMSR-E *fw* data and corresponding MOD44W results. The method helps to establish a stable relationship between the AMSR-E and MOD44W *fw* results without being

affected by the data distributions. Alternative methods, including CDF-matching, may produce similar or better results for aligning the *fw* data sets and will be examined in future work.

The relationships between AMSR-E and MODIS *fw* data are different at relatively small and relatively large AMSR-E *fw* conditions; the *fw*=0.15 threshold was selected for describing the different relationships as illustrated in the figure below:

[Figure]

Fig. 1 Relationships between AMSR-E ascending *fw* retrievals and MOD44W results for relatively small ($< 0.15$) (left) and relatively large ($\geq 0.15$) (right) AMSR-E *fw* conditions.

We added the following sentences to clarify the issue:
"… and (d) relationships between AMSR-E and MOD44W *fw* retrievals were analyzed based on their mean group values and derived for two respective conditions representing relatively different relationship slopes: AMSR-E *fw* <0.15 and *fw* ≥0.15. The 0.15 *fw* threshold was selected for describing the AMSR-E and MOD44W *fw* relationships over the different AMSR-E *fw* levels."

***Technical corrections***
***1. P4, L10: "(Du et al. 2015)" should be "Du et al. (2015)". Also see citations at P4, L28, and P8, L26.***

Reply: As suggested by the reviewer, the sentences were revised as follows:
"…described below, which follow from Du et al. (2015) but use different regression coefficients…";

"the iterative retrieval algorithm proposed in Jones et al. (2010) and revised in Du et al. (2015) assumes dry…"

"The LPDR *vsm* retrieval accuracy was evaluated using a similar approach as Du et al. (2016) by comparing the satellite"

***2. P10, L26: in "Tmn", "mn" should be subscript.***
Reply: As suggested by the reviewer, the sentence was revised as follows "…AMSR2 derived accuracies for $T_{mx}$ (RMSE = 3.64 ℃) and $T_{mn}$ (RMSE = 3.54 ℃) from a prior study…"